# Regret Bounds for Thompson Sampling in Episodic Restless Bandit Problems

**Young Hun Jung**
Department of Statistics
University of Michigan
yhjung@umich.edu

**Ambuj Tewari**
Department of Statistics
University of Michigan
tewaria@umich.edu

## Abstract

Restless bandit problems are instances of non-stationary multi-armed bandits. These problems have been studied well from the optimization perspective, where the goal is to efficiently find a near-optimal policy when system parameters are known. However, very few papers adopt a learning perspective, where the parameters are unknown. In this paper, we analyze the performance of Thompson sampling in episodic restless bandits with unknown parameters. We consider a general policy map to define our competitor and prove an $\tilde{\mathcal{O}}(\sqrt{T})$ Bayesian regret bound. Our competitor is flexible enough to represent various benchmarks including the best fixed action policy, the optimal policy, the Whittle index policy, or the myopic policy. We also present empirical results that support our theoretical findings.

## 1 Introduction

*Restless bandits* [Whittle, 1988] are variants of multi-armed bandit (MAB) problems [Robbins, 1952]. Unlike the classical MABs, the arms have non-stationary reward distributions. Specifically, we will focus on the class of restless bandits whose arms change their states based on Markov chains. Restless bandits are also distinguished from *rested bandits* where only the active arms evolve and the passive arms remain frozen. We will assume that each arm changes according to two different Markov chains depending on whether it is played or not. Because of their extra flexibility in modeling non-stationarity, restless bandits have been applied to practical problems such as dynamic channel access problems [Liu et al., 2011, 2013] and online recommendation systems [Meshram et al., 2017].

Due to the arms' non-stationary nature, playing the same set of arms for every round usually does not produce the optimal performance. This makes the optimal policy highly non-trivial, and Papadimitriou and Tsitsiklis [1999] show that it is generally PSPACE hard to identify the optimal policy for restless bandits. As a consequence, many researchers have been devoted to find an efficient way to approximate the optimal policy [Liu and Zhao, 2010, Meshram et al., 2018]. This line of work primarily focuses on the *optimization* perspective in that the system parameters are already known.

Since the true system parameters are unavailable in many cases, it becomes important to examine restless bandits from a *learning* perspective. Due to the learner's additional uncertainty, however, analyzing a learning algorithm in restless bandits is significantly challenging. Liu et al. [2011, 2013] and Tekin and Liu [2012] prove $\mathcal{O}(\log T)$ bounds for confidence bound based algorithms, but their competitor always selects a fixed set of actions, which is known to be weak (see Section 5 for an empirical example of the weakness of the best fixed action competitor). Dai et al. [2011, 2014] show $\mathcal{O}(\log T)$ bounds against the optimal policy, but their assumptions on the underlying model are very limited. Ortner et al. [2012] prove an $\tilde{\mathcal{O}}(\sqrt{T})$ bound in general restless bandits, but their algorithm is intractable in general.

In a different line of work, Osband et al. [2013] study Thompson sampling in the setting of a fully observable Markov decision process (MDP) and show the Bayesian regret bound of $\tilde{\mathcal{O}}(\sqrt{T})$ (hiding dependence on system parameters like state and action space size). Unfortunately, this result is not applicable in our setting as ours is partially observable due to bandit feedback. Following Ortner et al. [2012], it is possible to transform our setting to the fully observable case, but then we end up having exponentially many states, which restricts the practical utility of existing results.

In this work, we analyze Thompson sampling in restless bandits where the system resets at the end of every fixed-length episode and the rewards are binary. We emphasize that this episodic assumption simplifies our analysis as the problem boils down to a *finite time horizon* problem. This assumption can be arguably limited, but there are applications such as dynamic channel access problems where the channel provider might reset their system every night for a maintenance-related reason and the episodic assumption becomes natural. We directly tackle the partial observability and achieve a meaningful regret bound, which when restricted to the classical MABs matches the Thompson sampling result in that setting. We are not the first to analyze Thompson sampling in restless bandits, and Meshram et al. [2016] study this type of algorithm as well, but their regret analysis remains in the one-armed-case with a fixed reward of not pulling the arm. They explicitly mention that a regret analysis of Thompson sampling in the multi-armed case is an interesting open question.

## 2    Problem setting

We begin by introducing our setting. There are $K$ arms indexed by $k = 1, \cdots, K$, and the algorithm selects $N$ arms every round. We denote the learner's action at time $t$ by a binary vector $A_t \in \{0, 1\}^K$ where $||A_t||_1 = N$. We call the selected arms as *active* and the rest as *passive*. We assume each arm $k$ has binary states, $\{0, 1\}$, which evolve as a Markov chain with transition matrix either $P_k^{\text{active}}$ or $P_k^{\text{passive}}$, depending on whether the learner pulled the arm or not.

At round $t$, pulling an arm $k$ incurs a binary reward $X_{t,k}$, which is the arm's current state. As we are in the bandit setting, the learner only observes the rewards of active arms, which we denote by $X_{t,A_t}$, and does not observe the passive arms' rewards nor their states. This feature makes our setting to be a *partially observable Markov decision process*, or POMDP. We denote the history of the learner's actions and rewards up to time $t$ by $\mathcal{H}_t = (A_1, X_{1,A_1}, \cdots, A_t, X_{t,A_t})$.

We assume the system resets every episode of length $L$, which is also known to the learner. This means that at the beginning of each episode, the states of the arms are drawn from an initial distribution. The entire time horizon is denoted by $T$, and for simplicity, we assume it is a multiple of $L$, or $T = mL$.

### 2.1    Bayesian regret and competitor policy

Let $\theta \in \Theta$ denote the entire parameters of the system. It includes transition matrices $P^{\text{active}}$ and $P^{\text{passive}}$, and an initial distribution of each arm's state. The learner only knows the prior distribution of this parameter at the beginning and does not have access to the exact value.

In order to define a regret, we need a *competitor policy*, or a *benchmark*. We first define a class of deterministic policies and policy mappings.

**Definition 1.** *A deterministic policy $\pi$ takes time index and history $(t, \mathcal{H}_{t-1})$ as an input and outputs a fixed action $A_t = \pi(t, \mathcal{H}_{t-1})$. A deterministic policy mapping $\mu$ takes a system parameter $\theta$ as an input and outputs a deterministic policy $\pi = \mu(\theta)$.*

We fix a deterministic policy mapping $\mu$ and let our algorithm compete against a deterministic policy $\pi^\star = \mu(\theta^\star)$, where $\theta^\star$ represents the true system parameter, which is unknown to the learner.

We keep our competitor policy abstract mainly because we are in the non-stationary setting. Unlike the classical (stationary) MABs, pulling the same set of arms with the largest expected rewards is not necessarily optimal. Moreover, it is in general PSPACE hard to compute the optimal policy when $\theta^\star$ is given. Regarding these statements, we refer the readers to the book by Gittins et al. [1989]. As a consequence, researchers have identified conditions that the (efficient) myopic policy is optimal [Ahmad et al., 2009] or proven that a tractable index-based policy has a reasonable performance against the optimal policy [Liu and Zhao, 2010].

---
**Algorithm 1** Thompson sampling in restless bandits
---
1: **Input** prior $Q$, episode length $L$, policy mapping $\mu$
2: **Initialize** posterior $Q_1 = Q$, history $\mathcal{H} = \emptyset$
3: **for** episodes $l = 1, \cdots, m$ **do**
4:     Draw a parameter $\theta_l \sim Q_l$ and compute the policy $\pi_l = \mu(\theta_l)$
5:     Set $\mathcal{H}_0 = \emptyset$
6:     **for** $t = 1, \cdots, L$ **do**
7:         Select $N$ active arms $A_t = \pi_l(t, \mathcal{H}_{t-1})$
8:         Observe rewards $X_{t,A_t}$ and update $\mathcal{H}_t$
9:     **end for**
10:    Append $\mathcal{H}_L$ to $\mathcal{H}$ and update posterior distribution $Q_{l+1}$ using $\mathcal{H}$
11: **end for**
---

We observe that most of proposed policies including the optimal policy, the myopic policy, or the index-based policy are deterministic. Therefore, researchers can plug in whatever competitor policy of their choice, and our regret bound will apply as long as the chosen policy mapping is deterministic.

Before defining the regret, we introduce a *value function*

$$V_{\pi,i}^\theta(\mathcal{H}) = \mathbb{E}_{\theta,\pi}[\sum_{j=i}^{L} A_j \cdot X_j | \mathcal{H}]. \tag{1}$$

This is the expected reward of running a policy $\pi$ from round $i$ to $L$ where the system parameter is $\theta$ and the starting history is $\mathcal{H}$. Note that the benchmark policy $\pi^\star$ obtains $V_{\pi^\star,1}^{\theta^\star}(\emptyset)$ rewards per episode in expectation. Thus, we can define the regret as

$$R(T; \theta^\star) = mV_{\pi^\star,1}^{\theta^\star}(\emptyset) - \mathbb{E}_{\theta^\star} \sum_{t=1}^{T} A_t \cdot X_t. \tag{2}$$

If an algorithm chooses to fix a policy $\pi_l$ for the entire episode $l$, which is the case of our algorithm, then the regret can be written as

$$R(T; \theta^\star) = mV_{\pi^\star,1}^{\theta^\star}(\emptyset) - \mathbb{E}_{\theta^\star} \sum_{l=1}^{m} V_{\pi_l,1}^{\theta^\star}(\emptyset) = \mathbb{E}_{\theta^\star} \sum_{l=1}^{m} V_{\pi^\star,1}^{\theta^\star}(\emptyset) - V_{\pi_l,1}^{\theta^\star}(\emptyset).$$

We particularly focus on the case where $\theta^\star$ is a random and bound the following *Bayesian regret*,

$$BR(T) = \mathbb{E}_{\theta^\star \sim Q} R(T; \theta^\star),$$

where $Q$ is a prior distribution over the set of system parameters $\Theta$. We assume that the prior is known to the learner. We caution our readers that there is at least one other regret definition in the literature, which is called either *frequentist regret* or *worst-case regret*. For this type of regret, one views $\theta^\star$ as a fixed unknown object and directly bounds $R(T; \theta^\star)$. Even though our primary interest is to bound the Bayesian regret, we can establish a connection to the frequentist regret in the special case where the prior $Q$ has a finite support and the benchmark is the optimal policy (see Corollary 6).

## 3 Algorithm

Our algorithm is an instance of *Thompson sampling* or *posterior sampling*, first proposed by Thompson [1933]. At the beginning of episode $l$, the algorithm draws a system parameter $\theta_l$ from the posterior and plays $\pi_l = \mu(\theta_l)$ throughout the episode. Once an episode is over, it updates the posterior based on additional observations. Algorithm 1 describes the steps.

We want to point out that the history $\mathcal{H}$ fulfills two different purposes. One is to update the posterior $Q_l$, and the other is as an input to a policy $\pi_l$. For the latter, however, we do not need the entire history as the arms reset every episode. That is why we set $\mathcal{H}_0 = \emptyset$ (step 5) and feed $\mathcal{H}_{t-1}$ to $\pi_l$ (step 7). Furthermore, as we assume that the arms evolve based on Markov chains, the history $\mathcal{H}_{t-1}$ can be summarized as

$$(r_1, n_1, \cdots, r_K, n_K), \tag{3}$$

which means that an arm $k$ is played $n_k$ rounds ago and $r_k$ is the observed reward in that round. If an arm $k$ is never played in the episode, then $n_k$ becomes $t$, and $r_k$ becomes the expected reward from the initial distribution based on $\theta_l$. As we assume the episode length is fixed to be $L$, there are $L$ possible values for $n_k$. Due to the binary reward assumption, $r_k$ can take three values including the case where the arm $k$ is never played. From these, we can infer that there are $(3L)^K$ possible tuples of $(r_1, n_1, \cdots, r_K, n_K)$. By considering these tuples as states and following the reasoning of Ortner et al. [2012], one can view our POMDP as a fully observable MDP. Then one can use the existing algorithms for fully observable MDPs (e.g., Osband et al. [2013]), but the regret bounds easily become vacuous since the number of states depends exponentially on the number of arms $K$. Additionally, as we assumed a policy mapping, one might argue to use existing *expert learning* or classical MAB algorithms considering potential policies as experts or arms. This is possible, but the number of potential policies corresponds to the size of $\Theta$, which can be very large or even infinite. For this reason, existing algorithms are not efficient and/or their regret bounds become too loose.

Due to its generality, it is hard to analyze the time and space complexity of Algorithm 1. Two major steps are computing the policy (step 4) and updating posterior (step 10). Computing the policy depends on our choice of competitor mapping $\mu$. If the competitor policy has better performance but is harder to compute, then our regret bound gets more meaningful as the benchmark is stronger, but the running time gets longer. Regarding the posterior update, the computational burden depends on the choice of the prior $Q$ and its support. If there is a closed-form update, then the step is computationally cheap, but otherwise the burden increases with respect to the size of the support.

## 4 Regret bound

In this section, we bound the Bayesian regret of Algorithm 1 by $\tilde{\mathcal{O}}(\sqrt{T})$. A key idea in our analysis of Thompson sampling is that the distributions of $\theta^\star$ and $\theta_l$ are identical given the history up to the end of episode $l-1$ (e.g., see Lattimore and Szepesvári, Chp. 36). To state it more formally, let $\sigma(\mathcal{H})$ be the $\sigma$-algebra generated by the history $\mathcal{H}$. Then we call a random variable $X$ is $\sigma(\mathcal{H})$-*measurable*, or simply $\mathcal{H}$-*measurable*, if its value is deterministically known given the information $\sigma(\mathcal{H})$. Similarly, we call a random function $f$ is $\mathcal{H}$-measurable if its mapping is deterministically known given $\sigma(\mathcal{H})$. We record as a lemma an observation made by Russo and Van Roy [2014].

**Lemma 2. (Expectation identity)** *Suppose $\theta^\star$ and $\theta_l$ have the same distribution given $\mathcal{H}$. For any $\mathcal{H}$-measurable function $f$, we have*

$$\mathbb{E}[f(\theta^\star)|\mathcal{H}] = \mathbb{E}[f(\theta_l)|\mathcal{H}].$$

Recall that we assume the competitor mapping $\mu$ is deterministic. Furthermore, the value function $V_{\pi,i}^\theta(\emptyset)$ in (1) is deterministic given $\theta$ and $\pi$. This implies $\mathbb{E}[V_{\pi^\star,i}^{\theta^\star}(\emptyset)|\mathcal{H}] = \mathbb{E}[V_{\pi_l,i}^{\theta_l}(\emptyset)|\mathcal{H}]$, where $\mathcal{H}$ is the history up to the end of episode $l-1$. This observation leads to the following regret decomposition.

**Lemma 3. (Regret decomposition)** *The Bayesian regret of Algorithm 1 can be decomposed as*

$$BR(T) = \mathbb{E}_{\theta^\star \sim Q} \sum_{l=1}^{m} \mathbb{E}_{\theta_l \sim Q_l}[V_{\pi^\star,1}^{\theta^\star}(\emptyset) - V_{\pi_l,1}^{\theta^\star}(\emptyset)] = \mathbb{E}_{\theta^\star \sim Q} \sum_{l=1}^{m} \mathbb{E}_{\theta_l \sim Q_l}[V_{\pi_l,1}^{\theta_l}(\emptyset) - V_{\pi_l,1}^{\theta^\star}(\emptyset)].$$

*Proof.* The first equality is a simple rewriting of (2) because Algorithm 1 fixes a policy $\pi_l$ for the entire episode $l$. Then we apply Lemma 2 along with the tower rule to get

$$\mathbb{E}_{\theta^\star \sim Q} \sum_{l=1}^{m} \mathbb{E}_{\theta_l \sim Q_l} V_{\pi^\star,1}^{\theta^\star}(\emptyset) = \mathbb{E}_{\theta^\star \sim Q} \sum_{l=1}^{m} \mathbb{E}_{\theta_l \sim Q_l} V_{\pi_l,1}^{\theta_l}(\emptyset). \qquad \square$$

Note that we can compute $V_{\pi_l,1}^{\theta_l}(\emptyset)$ as we know $\theta_l$ and $\pi_l$. We can also infer the value of $V_{\pi_l,1}^{\theta^\star}(\emptyset)$ from the algorithm's observations. The main point of Lemma 3 is to rewrite the Bayesian regret using terms that are relatively easy to analyze.

Next, we define the *Bellman operator*

$$\mathcal{T}_\pi^\theta V(\mathcal{H}_{t-1}) = \mathbb{E}_{\theta,\pi}[A_t \cdot X_t + V(\mathcal{H}_t)|\mathcal{H}_{t-1}].$$

It is not hard to check that $V_{\pi,i}^\theta = \mathcal{T}_\pi^\theta V_{\pi,i+1}^\theta$. The next lemma further decomposes the regret.

**Lemma 4. (Per-episode regret decomposition)** *Fix $\theta^\star$ and $\theta_l$, and let $\mathcal{H}_0 = \emptyset$. Then we have*

$$V_{\pi_l,1}^{\theta_l}(\mathcal{H}_0) - V_{\pi_l,1}^{\theta^\star}(\mathcal{H}_0) = \mathbb{E}_{\theta^\star,\pi_l} \sum_{t=1}^{L} (\mathcal{T}_{\pi_l}^{\theta_l} - \mathcal{T}_{\pi_l}^{\theta^\star}) V_{\pi_l,t+1}^{\theta_l}(\mathcal{H}_{t-1}).$$

*Proof.* Using the relation $V_{\pi,i}^{\theta} = \mathcal{T}_{\pi}^{\theta} V_{\pi,i+1}^{\theta}$, we may write

$$V_{\pi_l,1}^{\theta_l}(\mathcal{H}_0) - V_{\pi_l,1}^{\theta^\star}(\mathcal{H}_0) = (\mathcal{T}_{\pi_l}^{\theta_l} V_{\pi_l,2}^{\theta_l} - \mathcal{T}_{\pi_l}^{\theta^\star} V_{\pi_l,2}^{\theta^\star})(\mathcal{H}_0)$$
$$= (\mathcal{T}_{\pi_l}^{\theta_l} - \mathcal{T}_{\pi_l}^{\theta^\star}) V_{\pi_l,2}^{\theta_l}(\mathcal{H}_0) + \mathcal{T}_{\pi_l}^{\theta^\star}(V_{\pi_l,2}^{\theta_l} - V_{\pi_l,2}^{\theta^\star})(\mathcal{H}_0).$$

The second term can be written as $\mathbb{E}_{\theta^\star,\pi_l}[(V_{\pi_l,2}^{\theta_l} - V_{\pi_l,2}^{\theta^\star})(\mathcal{H}_1)|\mathcal{H}_0]$, and we can repeat this $L$ times to obtain the desired equation. $\square$

Now we are ready to prove our main theorem. A complete proof can be found in Appendix A.

**Theorem 5. (Bayesian regret bound of the Thompson sampling)** *The Bayesian regret of Algorithm 1 satisfies the following bound*

$$BR(T) = \mathcal{O}(\sqrt{KL^3 N^3 T \log T}) = \mathcal{O}(\sqrt{mKL^4 N^3 \log(mL)}).$$

**Remark.** *If the system is the classical stationary MAB, then it corresponds to the case $L = 1, N = 1$, and our result reproduces the result of $\mathcal{O}(\sqrt{KT \log T})$ [Lattimore and Szepesvári, Chp. 36]. This suggests our bound is optimal in $K$ and $T$ up to a logarithmic factor. Further, when $N > \frac{K}{2}$, we can think of the problem as choosing the passive arms, and the smaller bound with $N$ replaced by $K - N$ would apply. When $L = 1$, the problem becomes combinatorial bandits of choosing $N$ active arms out of $K$. Cesa-Bianchi and Lugosi [2012] propose an algorithm with a regret bound $\mathcal{O}(\sqrt{KNT \log K})$ with an assumption that the loss is always bounded by 1. Since our reward can be as big as $N$, our bound has the same dependence on $N$ with theirs, suggesting tight dependence of our bound on $N$.*

*Proof Sketch.* We fix an episode $l$ and analyze the regret in this episode. Let $t_l = (l-1)L$ so that the episode starts at time $t_l + 1$. Define $N_l(k,r,n) = \sum_{t=1}^{t_l} \mathbb{1}\{A_{t,k} = 1, r_k = r, n_k = n\}$, which counts the number of rounds where the arm $k$ was chosen by the learner with history $r_k = r$ and $n_k = n$ (see (3) for definition). Note that $k \in [K], r \in \{0, 1, \rho(k)\}$, and $n \in [L]$, where $\rho(k)$ is the initial success rate of the arm $k$. This implies there are $3KL$ tuples of $(k, r, n)$.

Let $\omega^\theta(k,r,n)$ denote the conditional probability of $X_k = 1$ given a history $(r, n)$ and a system parameter $\theta$. Also let $\hat{\omega}(k,r,n)$ denote the empirical mean of this quantity (using $N_l(k,r,n)$ past observations and set the estimate to 0 if $N_l(k,r,n) = 0$). Then define

$$\Theta_l = \left\{ \theta \mid \forall (k,r,n), \ |(\hat{\omega} - \omega^\theta)(k,r,n)| < \sqrt{\frac{2\log(1/\delta)}{1 \vee N_l(k,r,n)}} \right\}.$$

Since $\hat{\omega}(k,r,n)$ is $\mathcal{H}_{t_l}$-measurable, so is the set $\Theta_l$. Using the Hoeffding inequality, one can show $\mathbb{P}(\theta^\star \notin \Theta_l) = \mathbb{P}(\theta_l \notin \Theta_l) \leq 3\delta KL$. In other words, we can claim that with high probability, $|\omega^{\theta_l}(k,r,n) - \omega^{\theta^\star}(k,r,n)|$ is small for all $(k,r,n)$.

We now turn our attention to the following Bellman operator

$$\mathcal{T}_{\pi_l}^{\theta} V_{\pi_l,t}^{\theta_l}(\mathcal{H}_{t-1}) = \mathbb{E}_{\theta,\pi_l}[A_{t_l+t} \cdot X_{t_l+t} + V_{\pi_l,t}^{\theta_l}(\mathcal{H}_t)|\mathcal{H}_{t-1}].$$

Since $\pi_l$ is a deterministic policy, $A_{t_l+t}$ is also deterministic given $\mathcal{H}_{t-1}$ and $\pi_l$. Let $(k_1, \ldots, k_N)$ be the active arms at time $t_l + t$ and write $\omega^\theta(k_i, r_{k_i}, n_{k_i}) = \omega_{\theta,i}$. Then we can rewrite

$$\mathcal{T}_{\pi_l}^{\theta} V_{\pi_l,t}^{\theta_l}(\mathcal{H}_{t-1}) = \sum_{i=1}^{N} \omega_{\theta,i} + \sum_{x \in \{0,1\}^N} P_x^\theta V_{\pi_l,t}^{\theta_l}(\mathcal{H}_{t-1} \cup (A_{t_l+t}, x)),$$

where $P_x^\theta = \prod_{i=1}^{N} \omega_{\theta,i}^{x_i}(1 - \omega_{\theta,i})^{1-x_i}$. Under the event that $\theta^\star, \theta_l \in \Theta_l$, we have

$$|\omega_{\theta_l,i} - \omega_{\theta^\star,i}| < 1 \wedge \sqrt{\frac{8\log(1/\delta)}{1 \vee N_l(k_i, r_{k_i}, n_{k_i})}} =: \Delta_i(t_l + t),$$

where the dependence on $t_l + t$ comes from the mapping from $i$ to $k_i$. When $\omega_{\theta_l,i}$ and $\omega_{\theta^\star,i}$ are close for all $(k,r,n)$, we can actually bound the difference between the following Bellman operators as

$$|(\mathcal{T}_{\pi_l}^{\theta^\star} - \mathcal{T}_{\pi_l}^{\theta_l})V_{\pi_l,t}^{\theta_l}(\mathcal{H}_{t-1})| \leq 3LN \sum_{i=1}^{N} \Delta_i(t_l + t).$$

Then by applying Lemma 4, we get $|V_{\pi_l,1}^{\theta_l}(\emptyset) - V_{\pi_l,1}^{\theta^\star}(\emptyset)| \leq 3LN\mathbb{E}_{\theta^\star,\pi_l} \sum_{t=1}^{L} \sum_{i=1}^{N} \Delta_i(t_l + t)$, which holds whenever $\theta^\star, \theta_l \in \Theta_l$. When $\theta^\star \notin \Theta_l$ or $\theta_l \notin \Theta_l$, which happens with probability less than $6\delta KL$, we have a trivial bound $|V_{\pi_l,1}^{\theta_l}(\emptyset) - V_{\pi_l,1}^{\theta^\star}(\emptyset)| \leq LN$. We can deduce

$$|V_{\pi_l,1}^{\theta_l}(\emptyset) - V_{\pi_l,1}^{\theta^\star}(\emptyset)| \leq 3LN\mathbb{1}(\theta^\star, \theta_l \in \Theta_l)\mathbb{E}_{\theta^\star,\pi_l} \sum_{t=1}^{L} \sum_{i=1}^{N} \Delta_i(t_l + t) + 6\delta KL^2 N.$$

Combining this with Lemma 3, we can show

$$BR(T) \leq 6\delta mKL^2 N + \mathbb{E}_{\theta^\star \sim Q} 3LN \sum_{l=1}^{m} \mathbb{1}(\theta^\star, \theta_l \in \Theta_l)\mathbb{E}_{\theta^\star,\pi_l} \sum_{t=1}^{L} \sum_{i=1}^{N} \Delta_i(t_l + t). \qquad (4)$$

After some algebra, bounding sums of finite differences by integrals, and applying the Cauchy-Schwartz inequality, we can bound the second summation by

$$18KL^3 N + 24\sqrt{3KL^3 N^3 T \log(1/\delta)}. \qquad (5)$$

Combining (4), (5), and our assumption that $T = mL$, we obtain

$$BR(T) = \mathcal{O}(\delta KLNT + KL^3 N + \sqrt{KL^3 N^3 T \log(1/\delta)}).$$

Since $NT$ is a trivial upper bound of $BR(T)$, we may ignore the $KL^3 N$ term. Setting $\delta = \frac{1}{T}$ completes the proof. $\square$

As discussed in Section 2, researchers sometimes pay more attention to the case where the true parameter $\theta^\star$ is deterministically fixed in advance, in which the frequentist regret becomes more relevant. It is not easy to directly extend our analysis to the frequentist regret in general, but we can achieve a meaningful bound with extra assumptions. Suppose our prior $Q$ is discrete and the competitor is the optimal policy. Then we know $R(T; \theta^\star)$ is always non-negative due to the optimality of the benchmark and can deduce $qR(T; \theta^\star) \leq BR(T)$, where $q$ is the probability mass on $\theta^\star$. This leads to the following corollary.

**Corollary 6. (Frequentist regret bound of Thompson sampling)** *Suppose the prior $Q$ is discrete and puts a non-zero mass on the parameter $\theta^\star$. Additionally, assume that the competitor policy is the optimal policy. Then Algorithm 1 satisfies the following bound*

$$R(T; \theta^\star) = \mathcal{O}(\sqrt{KL^3 N^3 T \log T}) = \mathcal{O}(\sqrt{mKL^4 N^3 \log(mL)}).$$

## 5 Experiments

We empirically investigate the Gilbert-Elliott channel model, which is studied by Liu and Zhao [2010] in a restless bandit perspective[1]. This model can be broadly used in communication systems such as cognitive radio networks, downlink scheduling in cellular systems, opportunistic transmission over fading channels, and resource-constrained jamming and anti-jamming.

Each arm $k$ has two parameters $p_{01}^k$ and $p_{11}^k$, which determine the transition matrix. We assume $P^{\text{active}} = P^{\text{passive}}$ and each arm's transition matrix is independent on the learner's action. There are only two states, *good* and *bad*, and the reward of playing an arm is 1 if its state is good and 0 otherwise. Figure 1 summarizes this model. We assume the initial distribution of an arm $k$ follows the stationary distribution. In other words, its initial state is good with probability $\omega_k = \frac{p_{01}^k}{p_{01}^k + 1 - p_{11}^k}$.

We fix $L = 50$ and $m = 30$. We use Monte Carlo simulation with size 100 or greater to approximate expectations. As each arm has two parameters, there are $2K$ parameters. For these, we set the prior distribution to be uniform over a finite support $\{0.1, 0.2, \cdots, 0.9\}$.

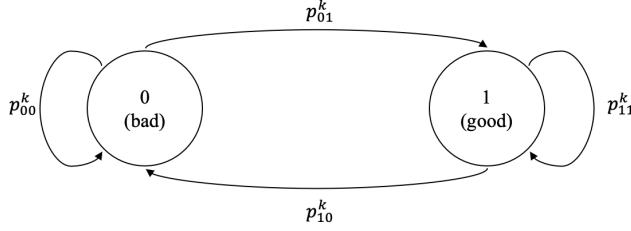

Figure 1: The Gilbert-Elliott channel model

## 5.1 Competitors

As mentioned earlier, one important strength of our result is that various policy mappings can be used as benchmarks. Here we test three different policies: the best fixed arm policy, the myopic policy, and the Whittle index policy. We want to emphasize again that these competitor policies know the system parameters while our algorithm does not.

The best fixed arm policy computes the stationary distribution $\omega_k = \frac{p_{01}^k}{p_{01}^k + 1 - p_{11}^k}$ for all $k$ and pulls the arms with top $N$ values. The myopic policy keeps updating the belief $\omega_k(t)$ for the arm $k$ being in a good state and pulls the top $N$ arms. Finally, the Whittle index policy computes the Whittle index of each arm and uses it to rank the arms. The Whittle index is proposed by Whittle [1988], and Liu and Zhao [2010] find a closed-form formula to compute the Whittle index in this particular setting. The Whittle index policy is very popular in optimization literature as it decouples the optimization process into $K$ independent problems for each arm, which significantly reduces the computational complexity while maintaining a reasonable performance against the optimal policy.

One observation is that these three policies are reduced to the best fixed arm policy in the stationary case. However, the first two policies are known to be sub-optimal in general [Gittins et al., 1989]. Liu and Zhao [2010] justify both theoretically and empirically the performance of the Whittle index policy for the Gilbert-Elliott channel model.

## 5.2 Results

We first analyze the Bayesian regret. For this, we use $K = 8$ and $N = 3$. The value functions $V_{\pi,1}^\theta(\emptyset)$ of the best fixed arm policy, the myopic policy, and the Whittle index policy are $105.4, 110.3$, and $111.4$, respectively. If a competitor policy has a weak performance, then Thompson sampling also uses this weak policy mapping to get a policy $\pi_l$ for the episode $l$. This implies that the regret does not necessarily become negative when the benchmark policy is weak. Figure 2 shows the trend of the Bayesian regret as a function of episode indices. Regardless of the choice of policy mapping, the regret is sub-linear, and the slope of log-log plot is less than $\frac{1}{2}$, which agrees with Theorem 5.

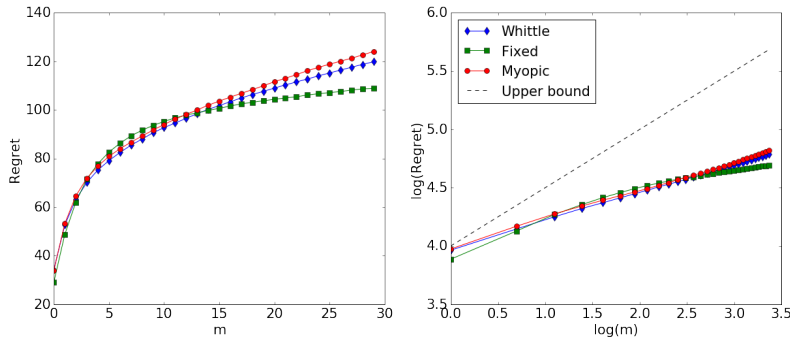

Figure 2: Bayesian regret of Thompson sampling versus episode (left) and its log-log plot (right)

Next we fix true parameters and investigate the model's behavior more closely. For this, we choose $K = 4$, $N = 2$, and $\{(p_{01}^k, p_{11}^k)\}_{k=1,2,3,4} = \{(0.3, 0.7), (0.4, 0.6), (0.5, 0.5), (0.6, 0.4)\}$. This

choice results in $\omega_k = 0.5$ for all $k$, and the best fixed arm policy becomes indifferent. Therefore achieving zero regret against the best fixed arm becomes trivial. We use the same uniform prior as the previous experiment. Figure 3 presents the trend of value functions and how Thompson sampling puts more posterior weights on the correct parameters as it proceeds. Three horizontal lines in the left figure represent the values of the competitor policies. The values of the best fixed arm policy, the myopic policy, and the Whittle index policy are $50.2, 54.6$, and $55.6$, respectively. It is a good example why one should not pull the same arms all the time in restless bandits. The value function of Thompson sampling successfully converges to the competitor value for every benchmark while the one with the myopic policy needs more episodes to fully converge. This supports Corollary 6 in that our model can be used even in the non-Bayesian setting as far as the prior has a non-zero weight on the true parameters. Also, the posterior weights on the correct parameters monotonically increase (Figure 3, right), which again confirms our model's performance. We measure these weights when the competitor map is the Whittle index policy.

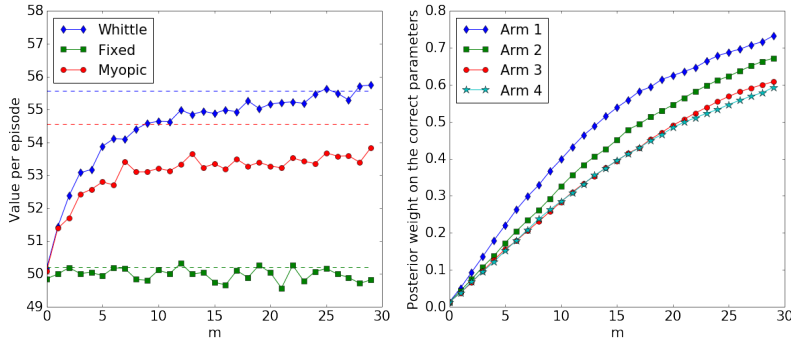

Figure 3: Average per-episode value versus episode and the benchmark values (left); the posterior weights of the correct parameters versus episode in the case of the Whittle index policy (right)

## 6 Discussion and future directions

In this paper, we have analyzed Thompson sampling in restless bandits with binary rewards. The Bayesian regret can be theoretically bounded as $\tilde{\mathcal{O}}(\sqrt{T})$, which naturally extends the results in the stationary MAB. One primary strength of our analysis is that the bound applies to arbitrary deterministic competitor policy mappings, which include the optimal policy and many other practical policies. Experiments with the simulated Gilbert-Elliott channel models support the theoretical results. In the special case where the prior has a discrete support and the benchmark is the optimal policy, our result extends to the frequentist regret, which is also supported by empirical results.

There are at least two interesting directions to be explored.

1. Our setting is episodic with known length $L$. The system resets periodically, which makes the analysis of the regret simpler. However, it is sometimes unrealistic to assume this periodic reset (e.g., online recommendation system studied by Meshram et al. [2017]). Analyzing a learning algorithm in the non-episodic setting will be useful.

2. Corollary 6 is not directly applicable in the case of continuous prior. In stationary MABs, it has been shown that Thompson sampling enjoys the frequentist regret bound of $\tilde{\mathcal{O}}(\sqrt{T})$ with additional assumptions [Lattimore and Szepesvári, Chp. 36]. Extending this to the restless bandit setting will be an interesting problem.

**Acknowledgments**

We acknowledge the support of NSF CAREER grant IIS-1452099. AT was also supported by a Sloan Research Fellowship. AT visited Criteo AI Lab, Paris and had discussions with Criteo researchers – Marc Abeille, Clément Calauzènes, and Jérémie Mary – regarding non-stationarity in bandit problems. These discussions were very helpful in attracting our attention to the regret analysis of restless bandit problems and the need for considering a variety of benchmark competitors when defining regret.

## Footnotes

[1]Our code is available at `https://github.com/yhjung88/ThompsonSamplinginRestlessBandits`

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
