[Supplementary Material · TSsupp.pdf]

# A  Proof of Theorem 5

We begin by introducing a technical lemma.

**Lemma 7.** *Let $a_i, b_i \in [0,1]$ and $|a_i - b_i| \le \Delta_i$ for $i \in [k]$. Then we can show*

$$\sum_{x \in \{0,1\}^k} |\prod_i a_i^{x_i}(1-a_i)^{1-x_i} - \prod_i b_i^{x_i}(1-b_i)^{1-x_i}| \le 2\sum_{j=1}^k \Delta_j. \tag{6}$$

*Proof.* Fix a binary vector $x$. For simplicity, let $c_i = a_i^{x_i}(1-a_i)^{1-x_i}$ and $d_i = b_i^{x_i}(1-b_i)^{1-x_i}$. Since $x_i$ is either 0 or 1, we have $|c_i - d_i| = |a_i - b_i| \le \Delta_i$. Then we can deduce

$$|\prod_{i=1}^k c_i - \prod_{i=1}^k d_i| \le (\prod_{i=1}^{k-1} c_i)|c_k - d_k| + |\prod_{i=1}^{k-1} c_i - \prod_{i=1}^{k-1} d_i|d_k$$

$$\le (\prod_{i=1}^{k-1} c_i)\Delta_k + |\prod_{i=1}^{k-1} c_i - \prod_{i=1}^{k-1} d_i|d_k$$

$$\le (\prod_{i=1}^{k-1} c_i)\Delta_k + (\prod_{i=1}^{k-2} c_i)\Delta_{k-1}d_k + |\prod_{i=1}^{k-2} c_i - \prod_{i=1}^{k-2} d_i|d_{k-1}d_k$$

$$\le \cdots$$

$$\le \sum_{j=1}^k (\prod_{i=1}^{j-1} c_i)\Delta_j(\prod_{i=j+1}^k d_i).$$

When summing up for all binary vectors $x$, we can write the coefficient of $\Delta_j$ as

$$\sum_{x \in \{0,1\}^k} (\prod_{i=1}^{j-1} c_i)(\prod_{i=j+1}^k d_i) = (\prod_{i=1}^{j-1} \sum_{x_i \in \{0,1\}} c_i)(\sum_{x_j \in \{0,1\}} 1)(\prod_{i=j+1}^k \sum_{x_i \in \{0,1\}} d_i)$$

$$= (\prod_{i=1}^{j-1} 1)2(\prod_{i=j+1}^k 1)$$

$$= 2,$$

where the second equality holds because $\sum_{x \in \{0,1\}} a^x(1-a)^{1-x} = a + (1-a) = 1$. This completes the proof. $\square$

Now we prove the main theorem.

**Theorem 5. (Bayesian regret bound of Thompson sampling)** *The Bayesian regret of Algorithm 1 satisfies the following bound*

$$BR(T) = \mathcal{O}(\sqrt{KL^3N^3T\log T}) = \mathcal{O}(\sqrt{mKL^4N^3\log(mL)}).$$

*Proof.* We fix an episode $l$ and analyze the regret in this episode. Let $t_l = (l-1)L$ so that the episode starts at time $t_l + 1$. Define

$$N_l(k, r, n) = \sum_{t=1}^{t_l} \mathbb{1}\{A_{t,k} = 1, r_k = r, n_k = n\}.$$

It counts the number of rounds where the arm $k$ was chosen by the learner with history $r_k = r$ and $n_k = n$ (see (3) for definition). Note that

$$k \in [K], r \in \{0, 1, \rho(k)\}, \text{ and } n \in [L],$$

where $\rho(k)$ is the initial success rate of the arm $k$. This implies there are $3KL$ tuples of $(k, r, n)$.

Let $\omega^\theta(k, r, n)$ denote the conditional probability of $X_k = 1$ given a history $(r, n)$ and a system parameter $\theta$. Also let $\hat{\omega}(k, r, n)$ denote the empirical mean of this quantity (using $N_l(k, r, n)$ past observations and set the estimate to 0 if $N_l(k, r, n) = 0$). Then define

$$\Theta_l = \{\theta \mid \forall(k, r, n), \ |(\hat{\omega} - \omega^\theta)(k, r, n)| < \sqrt{\frac{2\log(1/\delta)}{1 \vee N_l(k, r, n)}}\}.$$

Since $\hat{\omega}(k, r, n)$ is $\mathcal{H}_{t_l}$-measurable, so is the set $\Theta_l$. Using the Hoeffding inequality, one can show $\mathbb{P}(\theta^\star \notin \Theta_l) = \mathbb{P}(\theta_l \notin \Theta_l) \leq 3\delta KL$.

We now turn our attention to the following Bellman operator

$$\mathcal{T}_{\pi_l}^\theta V_{\pi_l, t}^{\theta_l}(\mathcal{H}_{t-1}) = \mathbb{E}_{\theta, \pi_l}[A_{t_l+t} \cdot X_{t_l+t} + V_{\pi_l, t}^{\theta_l}(\mathcal{H}_t)|\mathcal{H}_{t-1}].$$

Since $\pi_l$ is a deterministic policy, $A_{t_l+t}$ is also deterministic given $\mathcal{H}_{t-1}$ and $\pi_l$. Let $(k_1, \ldots, k_N)$ be the active arms at time $t_l + t$ and write $\omega^\theta(k_i, r_{k_i}, n_{k_i}) = \omega_{\theta, i}$. Then we can rewrite

$$\mathcal{T}_{\pi_l}^\theta V_{\pi_l, t}^{\theta_l}(\mathcal{H}_{t-1}) = \sum_{i=1}^{N} \omega_{\theta, i} + \sum_{x \in \{0,1\}^N} P_x^\theta V_{\pi_l, t}^{\theta_l}(\mathcal{H}_{t-1} \cup (A_{t_l+t}, x)), \tag{7}$$

where $P_x^\theta = \prod_{i=1}^{N} \omega_{\theta, i}^{x_i}(1 - \omega_{\theta, i})^{1-x_i}$. Under the event that $\theta^\star, \theta_l \in \Theta_l$, we have

$$|\omega_{\theta_l, i} - \omega_{\theta^\star, i}| < 1 \wedge \sqrt{\frac{8\log(1/\delta)}{1 \vee N_l(k_i, r_{k_i}, n_{k_i})}} =: \Delta_i(t_l + t), \tag{8}$$

where the dependence on $t_l + t$ comes from the mapping from $i$ to $k_i$. Lemma 7 provides

$$\sum_{x \in \{0,1\}^N} |P_x^{\theta_l} - P_x^{\theta^\star}| \leq 2\sum_{i=1}^{N} \Delta_i(t_l + t). \tag{9}$$

From (7), (9), and the fact that $|V_{\pi, t}^\theta| \leq LN$, we obtain given $\mathcal{H}_{t-1}$ and the event $\theta^\star, \theta_l \in \Theta_l$,

$$|(\mathcal{T}_{\pi_l}^{\theta^\star} - \mathcal{T}_{\pi_l}^{\theta_l})V_{\pi_l, t}^{\theta_l}(\mathcal{H}_{t-1})| \leq (2LN + 1)\sum_{i=1}^{N} \Delta_i(t_l + t) \leq 3LN\sum_{i=1}^{N} \Delta_i(t_l + t).$$

Then by applying Lemma 4, we get

$$|V_{\pi_l, 1}^{\theta_l}(\emptyset) - V_{\pi_l, 1}^{\theta^\star}(\emptyset)| \leq 3LN\mathbb{E}_{\theta^\star, \pi_l} \sum_{t=1}^{L} \sum_{i=1}^{N} \Delta_i(t_l + t).$$

The above inequality holds whenever $\theta^\star, \theta_l \in \Theta_l$. When $\theta^\star \notin \Theta_l$ or $\theta_l \notin \Theta_l$, which happens with probability less than $6\delta KL$, we have a trivial bound $|V_{\pi_l, 1}^{\theta_l}(\emptyset) - V_{\pi_l, 1}^{\theta^\star}(\emptyset)| \leq LN$. We can deduce

$$|V_{\pi_l, 1}^{\theta_l}(\emptyset) - V_{\pi_l, 1}^{\theta^\star}(\emptyset)| \leq 3LN\mathbb{1}(\theta^\star, \theta_l \in \Theta_l)\mathbb{E}_{\theta^\star, \pi_l} \sum_{t=1}^{L} \sum_{i=1}^{N} \Delta_i(t_l + t) + 6\delta KL^2 N.$$

Combining this with Lemma 3, we can show

$$BR(T) \leq 6\delta mKL^2 N + \mathbb{E}_{\theta^\star \sim Q} 3LN \sum_{l=1}^{m} \mathbb{1}(\theta^\star, \theta_l \in \Theta_l)\mathbb{E}_{\theta^\star, \pi_l} \sum_{t=1}^{L} \sum_{i=1}^{N} \Delta_i(t_l + t). \tag{10}$$

We further analyze the summation to finish the argument. Note that for this summation, we have $\theta^\star, \theta_l \in \Theta_l$. We shorten $N_l(k_i, r_{k_i}, n_{k_i})$ to $N_l$ for simplicity. By the definition of $\Delta_i$ in (8), we get

$$\sum_{l=1}^{m} \sum_{t=1}^{L} \sum_{i=1}^{N} \Delta_i(t_l + t) \leq \sum_{l=1}^{m} \sum_{t=1}^{L} \sum_{i=1}^{N} \mathbb{1}\{N_l \leq L\} + \Delta_i\mathbb{1}\{N_l > L\}$$

$$\leq 6KL^2 + \sum_{l=1}^{m} \sum_{t=1}^{L} \sum_{i=1}^{N} \mathbb{1}\{N_l > L\}\sqrt{\frac{8\log(1/\delta)}{N_l}}, \tag{11}$$

where the second inequality holds because there are $3KL$ possible tuples of $(k, r, n)$ and a tuple can contribute at most $2L$ to the first summation.

We can bound the second term as follows

$$
\begin{aligned}
\sum_{l=1}^{m}\sum_{t=1}^{L}\sum_{i=1}^{N} \mathbb{1}\{N_l > L\}\sqrt{\frac{1}{N_l}} &= \sum_{l=1}^{m}\sum_{(k,r,n)} \mathbb{1}\{N_l > L\}(N_{l+1} - N_l)\sqrt{\frac{1}{N_l}}\\
&\leq \sum_{l=1}^{m}\sum_{(k,r,n)} (N_{l+1} - N_l)\sqrt{\frac{2}{N_{l+1}}}\\
&\leq \sqrt{8}\sum_{(k,r,n)} \sqrt{N_{m+1}(k,r,n)}\\
&\leq \sqrt{24KLNT}.
\end{aligned}
\tag{12}
$$

For the first inequality, we use $N_{l+1} \leq N_l + L \leq 2N_l$. The second inequality holds due to the integral trick. Finally, the last inequality holds by the Cauchy-Schwartz inequality along with the fact that $\sum_{(k,r,n)} N_{m+1}(k,r,n) = NT$.

Combining (10), (11), (12), and our assumption that $T = mL$, we obtain

$$
BR(T) = \mathcal{O}(\delta KLNT + KL^3N + \sqrt{KL^3N^3T \log(1/\delta)}).
$$

Since $NT$ is a trivial upper bound of $BR(T)$, we may ignore the $KL^3N$ term. Setting $\delta = \frac{1}{T}$ completes the proof. $\square$