[Reviews · NeurIPS 2019]

Reviewer 1



The main problem with this article is that the mathematical formulation of the settings of the problem is not sufficiently precise to understand exactly what is done later in the paper. For example, the paper is not clear about what is called the "entire parameters of the systems". What is P^{active}, and P^{passive}? What is X_j in formula (1)? There were $X_{t,A_t}$ with two indexes, is it the same thing? What is the difference between a "true parameter" and a parameter... etc. The experimental part is unreadable: even with the maximum possible zoom, it remains very hard to read. Furthermore, several of the curves are plotted without true explanations on how they were produced precisely: e.g. how many runs. Why the curve on Figure 3 left, is like that? (jumps, no smoothness...)

Reviewer 2



Under the periodic and simultaneous restarting assumption, the Thompson sampling algorithm is quite straightforward. The significance is in the analysis of its regret. Allowing general benchmark policies in the regret analysis is a strength of the result. The writing is clear. My main concerns are as follows: 1. The simultaneous and periodic restart of all arms is rather limiting. Any motivating applications that give rise to this model? 2. Under such a restarting assumption, the problem is essentially reduced from a restless bandit problem to a problem with i.i.d. processes by viewing each epoch as a super-time-instant. The authors should discuss this view of the problem and its implications in regret analysis (e.g., difficulties in mapping analysis of Thompson sampling in i.i.d. reward processes to this setting under this view of i.i.d. super-time-instants, and how different the current analysis is from the existing ones on Thompson sampling for i.i.d. rewards). 3. The abstract and the introduction, especially the presentation and comparison with existing literature on restless bandits with unknown model, do not give a complete picture. The restrictive assumption of restarting (which also significantly simplifies the regret analysis) was not mentioned. Note that the work by Liu, et. al and by Tekin and Liu allow general Markov chain rather than two-state and do not need restarting assumption. The work by Dai, et. al, although require stochastically identical two-state arms, does not rely on the restarting assumption, either. A more complete picture of the result should be reflected in the title, abstract, and introduction. All these results, including this paper, make restrictive assumptions, largely due to the difficulty of the problem. They are just limited in different aspects. 3. Under known transition probabilities, with the restarting, the problem is a finite-horizon restless bandit problem with a horizon of length L (the period of restarting). In this case, the optimal policy is in general time-dependent. A more clear discussion of this issue and its ramifications in the regret analysis when the benchmark policy is the optimal policy will provide more insight and enhance the paper.

Reviewer 3



As I mentioned above, I consider the framework for Thompson sampling algorithms valuable, as it is easily implementable and allows for regret bounds with respect to a general class of policies. The analysis is also intuitive and not technically very difficult, combining ideas from posterior sampling that are now typically applied in Thompson sampling with characterizations of the Bayesian regret through the Bellman operator and concentration of parameter estimation. Proofs appear correct and formally written. One thing I want to point out is that the length of the episodes, L, necessarily needs to scale as o(T^{1/2}) (that is, sub-square-root of the horizon) to get a sublinear regret bound according to the statement of the main theorem of the paper. In the analysis, it appears as though the sample counts that are used to establish concentration of the parameter estimate in episode $l$ to the true parameter are based on the sample counts at the start of the episode -- these sample counts are not tracked through the episode itself. So the analysis seems to be tightest when $L$ (which is the length of the episode) is very small, say a constant and not even growing in $T$. My feeling is that considering the evolution of sample counts within episodes would make the analysis very delicate, but I am interested to hear from the authors about whether a more fine-grained analysis adds any value, and how difficult it may be to do. In particular, should I expect that the bounds in the main theorem are tight? Are lower bounds easy to prove? I ask these questions because I am curious about how fundamental the episodic setting is and in particular how the episodic length affects a regret bound. Are longer and fewer episodes necessarily much worse for regret in simulation, as is suggested in the theoretical upper bound? I already consider the results in this submission valuable, given that the restless multi-armed bandit problem is already very difficult and this is the first analysis involving learning the transition matrices that I have seen -- but concrete answers to these questions would make me consider raising my score higher.

[Author Response · NeurIPS 2019]

We thank all reviewers for their comments. Minor comments will be addressed in the final version.

## Reviewer 1

**Clear description of the setting** We want to emphasize first that our problem setting is a standard restless bandit
setting with a few specific choices. $P_k^{\text{active}}$ and $P_k^{\text{passive}}$ are the transition matrices of the arm $k$ when it is pulled or not,
respectively. $X_t$ is a $K$ dimensional binary vector such that the $k^{th}$ component $X_{t,k}$ represents the reward of the arm
$k$. Since the learner only observes the rewards of pulled arms, only the $N$ components $X_{t,A_t}$ will be available to the
learner. These notions are defined in lines 49 - 58. We will make our description clearer in the final version.

**Messages of the experiments** The first experiment empirically checks the Bayesian regret of our algorithm is indeed
$\tilde{O}(\sqrt{T})$. The second experiment shows the algorithm still works in the frequentist setting. Figure 3 (left) illustrates
how the value function of the policy $\pi_l$ chosen in an episode converges to the baseline value for a variety of competitor
mappings (the best fixed action, the myopic policy, and the Whittle index policy). Figure 3 (right) shows the posterior
weights on the true parameters monotonically increase. We will describe the details of our experiments more carefully
and make figures more readable.

## Reviewer 2

**1. Motivating application of the episodic setting** Yes, the assumption of periodic restart of the system is somewhat
limiting, and the regret analysis in the infinite time horizon is an interesting open question. Analyzing the finite
horizon case should be an intermediate step towards solving this question. Moreover, the episodic case itself has a few
motivating applications. For example, in the dynamic channel access problem that we consider in our experiment, the
channel provider might reset their system every night when network traffic is low for maintenance related reasons. After
the reset, every channel should be available for use, which can be thought as the beginning of a new episode.

**2. Super-time-instant** It is indeed possible to tackle the problem by considering each deterministic policy as an arm.
However, this would result in very large (possibly infinite) $K$, the number of arms, and the existing bounds become
vacuous as they depend (polynomially) on $K$. The bound in Dai et al. [2011] is meaningful since there are only two
competing policies. This perspective still conveys interesting points, and we will add the comparison in the final version.

**3. More complete picture in intro** We totally agree that existing results, including ours, are just limited in different
aspects. We will clarify this point more clearly. Nevertheless, we want to point out that this is the *first* paper that
analyzes Thompson sampling in *multi-armed* restless bandit problems.

**4. The optimal policy depends on $L$** Yes, the optimal policy will depend on the episode length. It will change the
baseline value in our regret definition in (2), but the same regret bound will still apply. It is one of our main contributions
that the regret bound applies regardless of the choice of the benchmark.

## Reviewer 3

**Finer analysis within each episode** First of all, the point raised by the reviewer is completely true. The episode length
$L$ should remain small to make the regret bound meaningful. We mainly considered the case where $L$ is fixed as a
constant and the number of resets, $m$, increases arbitrarily so that the posterior distribution concentrates sufficiently
around the truth. A fundamental reason why we did not do the finer analysis within the episode is because our algorithm
fixes a policy $\pi_l$ and runs it throughout the episode $l$. If we get to do the finer analysis, then that means our algorithm
changes the policy more often, which comes with an extra cost. For example, in the regret analysis by Ouyang et
al. ["Learning Unknown Markov Decision Processes: A Thompson Sampling Approach," NIPS 2017], who analyze
Thompson sampling in non-episodic fully observable MDPs, the bound includes $K_T$, the number of different policies
that Thompson sampling runs up to time $T$.

**Tightness of the regret bound** As pointed out in the remark right after Theorem 5, our result reproduces the regret
bound of $O(\sqrt{KT \log T})$ in the stationary MAB problem, whose lower bound is shown to be $\Omega(\sqrt{KT})$. This suggests
that our bound is optimal in $K$ and $T$ up to a logarithmic factor. When $L = 1$, the problem becomes a combinatorial
bandit problem (of choosing a set of $N$ active arms out of a total of $K$) in which case the best known regret bound is
$O(\sqrt{KN^3T \log K})$ (e.g., see "Combinatorial bandits" by Cesa-Bianchi and Lugosi [2012]. Their bound is actually
$O(\sqrt{KNT \log K})$, but they normalize the loss to be in $[0, 1]$, whereas our reward is in $[0, N]$). Our bound agrees with
their bound up to logarithmic terms. Finally, the optimal dependence on $L$ remains open. We will add the discussion of
tight dependence in the final version.

[Meta-Review · NeurIPS 2019]

The reviewers liked this paper, and I did as well. One thought is whether or not Exp4 can be adapted to this setting. The translation is not immediate by any means, but perhaps this is worth thinking about. Please take the reviewers suggestions into consideration for the final version as promised in your response.